# Fungal Strain Influences Thermal Conductivity, Hydrophobicity, Color Homogeneity, and Mold Contamination of Mycelial Composites

**DOI:** 10.3390/ma17246050

**Published:** 2024-12-11

**Authors:** Joris Verhelst, Simon Vandersanden, Olivier Nouwen, François Rineau

**Affiliations:** Environmental Biology, Centre for Environmental Sciences, Hasselt University, 3590 Diepenbeek, Belgium; joris.verhelst03@gmail.com (J.V.); simon.vandersanden@uhasselt.be (S.V.); olivier.nouwen@uhasselt.be (O.N.)

**Keywords:** mycelial composites, mycomaterials, thermal conductivity, hydrophobicity, color homogeneity, mold contamination

## Abstract

Mycomaterials are biomaterials made by inoculating a lignocellulosic substrate with a fungus, where the mycelium acts as a binder and enhances material properties. These materials are well suited as sustainable alternatives to conventional insulation materials thanks to their good insulation properties, low density, degradability, and fire resistance. However, they suffer from mold contamination in moist environments and poor perception (“organic” appearance). Furthermore, most mycomaterials to date have been derived from a limited range of fungal species, leaving the vast phenotypic diversity of fungi largely untapped. We hypothesized that by exploring a broader range of strains, we could enhance the likelihood of discovering a material that meets the needs for insulation panels. We generated mycomaterials from nine fungal strains and measured their thermal conductivity, mold resistance, and perception properties. We observed significant variations across strains on these three parameters. Thermal conductivity ranged from levels comparable to extruded polystyrene to nearly as effective as polyurethane (0.039 to 0.019 W/mK). All materials generated were hydrophobic (equivalent to 105–122° contact angle), but differed by a factor of two in color appearance and sensitivity to mold (0–94% of surface colonized). We also found a method to improve resistance to mold using deactivated contaminant propagules.

## 1. Introduction

Thermal insulation is of great importance to reduce our carbon footprint. In 2021, the share of home heating in the residential construction sector in Europe was 64%. The building sector globally emits 39% of CO_2_ emissions and uses 36% of the world’s energy [1]. Therefore, finding ways to reduce the carbon footprint of households and the building sector is vital to improve our environmental impact.

Thermal insulation is made of material that retards the rate of heat flow by conduction, convection, and radiation [2]. Thermal conductivity is used to quantify insulation performance, and is defined as the heat flow that passes through a unit area of a 1 m thick homogeneous material due to a temperature gradient equal to 1 K, expressed in W/m/K [3]. A lower value for thermal conductivity means that less heat is transferred, and therefore better insulation. Conventional insulation materials (mineral wools, expanded or extruded polystyrene, or polyisocyanate) offer good thermal conductivity, but their production processes are energy-intensive, rely on non-renewable resources, and often generate harmful byproducts. Recycling these materials also proves challenging, leading to significant environmental impacts [4,5,6,7,8,9].

Renewable insulation materials like hemp, kenaf, cellulose, and flax offer thermal conductivity comparable to synthetic insulators while storing carbon, requiring less production energy, and having a lower environmental impact. However, they easily absorb water and often rely on non-sustainable additives like fire retardants, binders, and fungicides [1], which can comprise up to 15% of their dry weight and include non-degradable compounds such as borax, urea formaldehyde, and polyesters. Some of these issues may be addressed by another class of materials called mycomaterials. These are bio-based materials based on fungal mycelium. They have a lower energy footprint and reliance on fossil fuels than conventional insulation materials [10]. Mycomaterials are used in a number of applications, but for heat insulation they are usually prepared as composites of mycelium growing on a lignocellulosic substrate. However, recent LCA assessments of mycomaterials have shown that their environmental footprint may not be as good as speculated, mostly for providing adequate growing and curation temperatures [11,12]. Mycelium, the vegetative structure of fungi, consists of a complex, branching 3D network of interwoven, elongated hyphal cells that penetrate the substrate. As it grows, the fungus digests and alters the substrate’s chemical composition while binding its fibers together. Both the lignocellulosic substrate and the mycelium contribute to their insulation properties. The fibers in lignocellulosic substrates resist heat flow by convection due to the low porosity between their fibers. This impedes air flow and heat transport [13]. Due to the friction of their fibers against fluid (air) flow, convective heat flow is strongly reduced. Untreated lignocellulosic substrates, including low-grade organic and agricultural waste, are ideal for mycomaterial production. Among these, hemp stands out due to its robust, stiff fibers [13] and versatile applications in textiles, pharmaceuticals, and construction, particularly as a wood substitute or insulating material [14]. Hemp shives, the woody core of hemp stems, are already used as loose-fill insulation with a thermal conductivity of 0.049–0.052 W/(m·K) and a minimal density of 109 kg/m^3^, comparable to XPS and glass wool. Hemp’s rapid growth, high carbon storage capacity, and prior success as a substrate for mycomaterials [15,16] make it an excellent choice for sustainable biomaterial production.

Selecting the appropriate substrate is essential for producing mycomaterial insulation panels, but it is the mycelium that distinguishes mycelial composites from other materials. Fungal hyphae, which form the mycelial network, are primarily composed of chitin (10–20% dry weight) and beta-glucans (50–60% dry weight) [17]. Mycelial colonization significantly enhances the physico-chemical properties of the material [18]. In terms of mechanical properties, mycomaterials exhibit compressive strength comparable to or better than XPS [19] and superior to EPS [20]. However, they typically have higher density. Flexural strength is also better than that of synthetic insulation materials [12,21]. Research suggests that the mechanical properties are more affected by fiber condition, size, and processing than by their chemical composition [16]. Thermal resistance studies on composites using Miscanthus as a substrate show thermal conductivity values ranging from 0.0822 to 0.104 W/(m·K) [22]. Mycelial composites made from hemp and *Trametes versicolor* have also proven suitable for insulation [16]. While prolonged mycelial colonization can improve thermal conductivity by decreasing density and increasing air content, choosing a substrate with inherently lower thermal conductivity [23], such as hemp or straw [12], is a more efficient strategy [23].

While research has highlighted the potential of mycomaterials for thermal insulation, it has also identified significant challenges that need to be addressed. First, they can absorb up to 200% of their weight in water when submerged for 48 h [24]. This negatively impacts thermal resistance because of the high thermal conductivity of water [25]. Fungal skin (dense mycelium produced by certain species when growing at exposed surfaces) can reduce the water absorption by providing a hydrophobic barrier to water uptake [26]. Second, upscaling the production process and producing consistent materials is challenging, as it takes a minimum of 2 weeks for the materials to be finished [21]. Third, mycomaterials, being biodegradable, are also susceptible to decomposition by mold contaminants in moist environments, which can lead to loss of the properties of the material. This is especially an issue for application as insulation materials, since those are constantly exposed to temperature fluctuations causing condensation. This creates an ideal environment for mold growth. Finally, despite their potential as biomaterials, mycomaterials face challenges in public acceptance, largely due to their inconsistent color and texture [27].

Most mycomaterial experiments rely on a limited number of model fungal species, leaving the vast diversity of these organisms largely untapped. Indeed, in 2019, only 36 of the estimated 2–11 million species have been reported in mycomaterial-related patents [28]. There are between 2–11 million fungal species [29]. This diversity encompasses a wide variation of traits that can influence the formation of a mycomaterial, such as enzyme production profile, hyphal types, color, hydrophobicity, or density, which can in turn be affected by the substrate they are growing on.

In this study, our goal was to explore a broad range of not commonly used fungal strains, to increase our potential to reveal significant variations in properties relevant for thermal insulation materials. We did not only measure thermal conductivity, but also hydrophobicity and resistance to mold contamination, as those are common issues with biomaterials exposed to many condensation cycles, as well as color homogeneity, as it is essential for their societal acceptance. For this, we isolated and screened nine wild fungal strains and generated insulation material composite samples by growing them on hemp shives. Our hypothesis was that mold resistance of mycelial composites could be improved by natural means (species selection and exposure to mold propagules during growth). Our results showed that some of the materials we prepared had excellent thermal insulation properties, as well as how color homogeneity and hydrophobicity are positively correlated. We also identified a new method to improve resistance to mold contamination.

## 2. Materials and Methods

### 2.1. Isolation of Wild Fungal Strains

Fungal sporocarps were collected and morphologically identified from Thier de Statte, Spa, Belgium (50.47398, 5.8785) at 23 August 2023, from Diepenbeek (50.92467188679654, 5.402792780728566) at 30 October 2023 and when encountered throughout the year in Diepenbeek and Halen. Isolation of the fungi was carried out under sterile conditions. After breaking open the sporocarp, a small piece of dikaryotic mycelium was cut with a scalpel or plucked with a tweezer and placed on a petri dish with malt extract agar (MEA) or potato dextrose agar (PDA) supplemented with streptomycin. In first instance, fungal species were identified morphologically.

The identity of four of these morphotypes (*Daedaleopsis tricolor*, *Stereum hirsutum*, *Fomes fomentarius A*, and *Trametes versicolor*) was confirmed by Sanger sequencing. To extract DNA, fungal biomass was generated by scraping mycelium pieces of colonized agar plates and adding them to liquid MEA (4 mL) in Eppendorf tubes incubated at 30 °C on a shaker (70 rpm) for one week. DNA was isolated using the Qiagen DNAeasy PowerSoil isolation kit (Qiagen, Hilden, Germany). We used nanodrop and Qbit to assess DNA quality and quantity. Subsequent PCR was performed using the Roche Applied Science “FastStart High Fidelity PCR System” (Roche Applied Science, Penzberg, Germany). We used primer set ITS1F-ITS4 (targeting the ITS1, 5.8srRNA, and ITS2 region of the fungal region) and ITS86F-ITS4 (targeting the 5.8s RNA and ITS2 region of the fungal ITS), depending on the species. Annealing temperatures used for the two primers pairs were 55 and 57 °C, respectively.

### 2.2. Preparation of Composite Materials

Initial substrate inoculation was carried out using grain spawn. Wheat grains were soaked for 24 h in distilled water, optionally cooked for 10 min after soaking, and then autoclaved. Grains not directly used were stored in the fridge and re-autoclaved before use if stored longer than a month. Inoculation of the grains was carried out by placing a colonized piece of agar (1 cm × 1 cm) in a petri dish filled with grains. This petri dish was incubated at 25 °C until fully colonized.

Hemp shives were acquired from Happy Home (Happy Home Hennepvezel—bodembedekking 100% natuurlijk, Buying4pets B.V., Ede, The Netherlands). Half of the hemp shives were ground to a particle size of 0.5 cm. The dried fibers were then weighed, hydrated with distilled water, homogenized, and autoclaved. The fiber composition comprised 50% ground fibers and 50% unground fibers.

Inoculation of fibers was carried out under sterile conditions. The following procedure were used: 14.4 g of dry fibers were hydrated with 43.2 mL distilled water and inoculated with 1.6 g of grain spawn on top of the fibers (method A) inside growth containers O95/40 + OD95 (cylindrical, 80 mm diameter, 40 mm height) with #40 green filter (SacO_2_, Deinze, Belgium). Other materials were made by filling petri dishes with 10.8 g of hydrated (75% moisture content), autoclaved fibers and adding 1.2 g of grain spawn. Finally, the surface was lightly pressed with a sterilized material.

After incubation, raw mycelial composites were removed from their growth container (Method A). To do this, a spatula was used to loosen the sides of the composites from the container and the material was lifted from underneath. Samples grown in petri dishes (Method B) were left in place. Then materials were dried for 72 h at 70 °C (Method A) or left in opened petri dishes for 5 h at 70 °C (Method B).

The preparation of the composite materials is summarized in Appendix A.

### 2.3. Measurement of Thermal Conductivity

We designed the following test to evaluate the thermal conductivity of the materials. The same samples as those used for the hydrophobicity and color evenness tests were used. The principle was to measure the temperature increase inside of an insulated box, which would be closed with a lid containing the tested material, exposed to 60 °C for 30 min (see Appendix A). Materials with a better insulation performance reduce the amount of heat transferred to the inside of the box, leading to a smaller temperature increase. The insulated box (inside dimensions: 17 cm × 17 cm × 17 cm) was made from wood panels on the outside and insulated from the interior with 6 cm thick XPS. The lid, also made of 6 cm thick XPS, was cut in two halves with a hole in the middle matching the dimensions of the tested materials (r = 3.5 cm). Sample materials were placed in this lid hole, and the contact surface between the material from the lid and the samples was sealed with insulation tape (see Appendix A). We placed a temperature sensor inside the box (SensorBlue Brifit hygrometer, DE-PARENT-WA59, Shanghai, China) to follow the temperature change. The box with lid and material was then placed into a preheated oven at 60 °C. Boxes were cooled down between measurements at room temperature for 30 min. Before measurements began, the box with the lid (but with no samples mounted) was placed in the oven for 30 min. XPS was used for benchmarking: we cut pieces of XPS foam with the same shape as the mycomaterial samples, and made a standard curve of thermal conductivity of either 1, 2, and 3 cm or 2, 3, and 4 cm thick XPS samples. This standard curve was repeated four times on four separate days to eventually correct for changes in room temperature: rho = −0.999 and *p* = 0.003713 (1308), rho = −1 and *p* < 2.2 × 10^−16^, rho = −0.993 (1408) and *p* = 0.07517 (1908), rho = −0.9975 and *p* = 0.04373 (1805) (Appendix A). The temperature difference before and after 30 min in the preheated oven were calculated. We used the XPS standard curve to calculate the thickness of XPS insulation material needed to achieve the same temperature difference as the tested mycelial composite. This value was then divided by the thickness of the mycelial composite to find the equivalent thickness of XPS needed to achieve the same thermal conductivity as 1 cm of the mycelial composite.

### 2.4. Measuring Pigmentation

Color measurement was carried out on the same samples as the hydrophobicity measurements. The materials were placed in a white box and images of the upper surface were taken with a Google pixel 4a smartphone camera, using flash, in a windowless room. The pictures were then cut out digitally in Krita. Empty pixels were filled in with a bright red color when exporting as a .jpeg file. The pictures were then processed using ImageJ version 1.54g. Color thresholding was used to calculate the total surface area of the material by selecting and counting all pixels darker than the bright red background. The settings for the color threshold menu were as follows: thresholding method “default” threshold color “red”, color space “RGB”, and the dark background box unchecked. The red color channel had values 0 to 166 and the “pass” box unchecked. This selects all red colors darker than the bright red background. For the blue color channel, all values between 0 and 135 passed the threshold, and in the green color section all color values between zero and 139 passed. We pressed select and then measure to measure all of the bright red pixels. This measurement was subtracted from the total amount of pixels in the image (calculated by multiplying the width of the picture with its length) to obtain the number of pixels from the material. After this, the images were converted to 8-bit grayscale and thresholded two times consecutively from 0 to 85 in “default” and “red” mode, where all values lighter than the grayscale-converted bright red background color were selected. Only the box with “don’t reset range” was selected. The surface area was then calculated by using the rectangle selection tool to select the entire image. The “measure” function was used with the settings configured to measure “area” and “limit to threshold” (Appendix A). Finally, the white surface area was divided by the total surface area of the material to determine the ratio of white surface area to the total surface area. This gave us the colorless ratio, a value for material colorlessness. The colorless ratio was measured relative to bright red because that was a color not occurring in the mycelial composites. Also, when there was discoloration, it was dark red to brown colored. Dark red to brown colored spots are darker than bright red when converting to grayscale, and would thus be filtered out when selecting even colored zones (Appendix A).

### 2.5. Measuring Hydrophobicity

*D. tricolor*, *S. hirsutum*, *Bjerkandera adusta*, *Ganoderma applanatum*, *Ganoderma lucidum*, *Ganoderma spec.*, *Phellinus nigricans*, *Hericium erinaceus*, *F. fomentarius*, and *T. versicolor* (M99) were inoculated on hemp using method A. After three weeks of growth at 23 °C, materials were oven dried and hydrophobicity was measured with the alcohol percentage test (APT) described by [30]. First, a dilution series with a step of 1% of ethanol in water was prepared. Ethanol droplets (8 µL) were applied on the surface of the material and the dilution where the droplet disappeared within 5 s was noted; the higher the dilution, the more hydrophobic the surface. This was carried out in five separate places for each material. For pigmented materials, both pigmented and non-pigmented parts of the material were tested in ratios that respected the surface ratio of the material that was considered pigmented. Homogenous materials were tested radially from the middle to the side. The hydrophobicity of each material was calculated as the mean value of all these measurements.

### 2.6. Measuring Contamination Resistance

Using method B, *D. tricolor*, *S. hirsutum*, *B. adusta*, *G. applanatum*, *G. lucidum*, *G. spec.*, *P. ignarius*, *H. erinaceus,* and *T. versicolor* (M99) were inoculated on hemp without added glucose (experiment A). For practical reasons, the other strains (*T. versicolor*, *D. tricolor*, and *S. hirsutum*) had to be ran on another batch for which grain spawn was not available, and hence 1% glucose was added to the substrate to compensate for the lack of C resources present in the grain (experiment B). For this reason, the results from experiments A and B will be discussed separately. This was carried out ten times for each strain. Samples were incubated at 23 °C for 24 days. On day 25, the surface of the sample was covered with cellophane containing a 1 cm^2^ cutout in the middle. The fungal skin underneath this cutout was opened with sterilized tweezers. On five of the ten samples (treatment group), an autoclaved *Trichoderma* spore suspension (1.6 × 10^5^ spores on the 1 cm^2^ cutout) was applied on days 25, 27, and 29 after inoculation. On the five other samples (control group), an equivalent volume of autoclaved distilled water was used instead. On day 32, all ten samples were dried for five hours at 70 °C. After drying, all materials were moisturized under sterile conditions by spraying with autoclaved distilled water and inoculated with a viable *Trichoderma* spore suspension on the treated square surface, using the same spore load (1.6 × 10^5^ spores/cm^2^) as the treatment. All samples were incubated at 30 °C and 90% relative humidity in filter boxes (same SacO_2_ boxes as above). A near-saturation moisture level in the headspace of the filter boxes was achieved with a saturated salt solution of Na_2_SO_4_. On day 6, after inoculation of the *Trichoderma*, pictures were taken of the top of the samples. From these pictures, the surface area of the sample that was colonized by the mold was quantified with ImageJ as the ratio of green (*Trichoderma* spores) surface area/total surface area. Marking the surface area covered by *Trichoderma* was carried out manually with the polygonal selection tool.

### 2.7. Statistics

We checked for normality using the Shapiro–Wilk test for all experiments. When ANOVA was used, linearity and homoscedasticity were controlled visually. For contamination resistance, we tested the effect of *Trichoderma* spore pretreatment (3 levels: uncolonized fibers, *Trichoderma*-treated, and *Trichoderma*-untreated) on the proportion of the surface covered with *Trichoderma* using a Kruskall–Wallis test (separately for each strain). Dunn’s tests were used to compare the mean area of surface covered in *Trichoderma* spores between conditions, and when a significant difference was detected between CO, C, and/or T, further analysis was performed using the Mann–Whitney-Wilcoxon test to determine which condition was significantly larger. The Kruskall–Wallis test was used to determine the effect of fungal strain on the ratio *Trichoderma* in the C and T treatment for experiment A and B separately. Dunn’s test was used for post hoc analysis. The effect of fungal strain (independent variable) on hydrophobicity, as the mean percentage dilution of ethanol that was absorbed by the material surface (dependent variable), was determined using ANOVA. Duncan’s new multiple range test was used for post hoc analysis. Differences in color evenness (dependent variable) for each fungal strain were analyzed using the Kruskall–Wallis test, and post hoc analysis was carried out using Dunn’s test. In the insulation test, the statistical analysis was performed by fitting a linear regression model on the temperature difference measured (dependent variable) as a function of the thickness of XPS. Validation of this test was performed by investigating the Spearman’s correlation between temperature difference and thickness. Comparing the mean thermal conductivity of samples to 1 was carried out by t-test (for normally distributed data) or Mann–Whitney–Wilcoxon tests. Investigating the influence of fungal strain on thermal conductivity was carried out with the Kruskall–Wallis test. Dunn’s test was used for post hoc analysis. Statistical analyses were performed using the R statistical software, version 4.2.2 [31]. 

## 3. Results

### 3.1. Thermal Conductivity

The purpose of this measurement was to assess the thermal conductivity of mycocomposites produced with nine different fungal strains on a hemp substrate and to compare their performance against XPS (extruded polystyrene) as a benchmark.

Figure 1 displays the thickness of XPS needed to achieve the same thermal conductivity in this test as 1 cm of the tested mycomaterial. Significant differences from one (XPS equivalent) were observed when testing for greater performance of the mycomaterial for UHM (*t*-test, *p* = 0.0363), HE (*t*-test, *p* = 0.0303), and BA (Mann–Whitney test, *p* = 0.0039). No condition performed significantly worse than the XPS. The thermal conductivity did not differ significantly between strains. Pearson’s correlation between thickness and temperature differences for the PUR control was calculated four times on separate measurements performed on four separate days: rho = −0.999 and *p* = 0.003713, rho = −1 and *p* < 2.2 *×* 10^−16^, rho = −0.993 and *p* = 0.07517, rho = −0.9975 and *p* = 0.04373 (Appendix A). The minimal, maximum, and standard errors of the cm XPS/cm material for all strains are displayed in Appendix A.

### 3.2. Pigmentation

The colorless ratio integrated how homogenous the colors were at the surface of the mycomaterial, as we expect that homogenous coloration is perceived better. It was measured as the ratio of the material that is lighter than bright red after conversion to the 8-bit grayscale. Figure 2 shows color evenness of mycomaterials made for nine separate strains. The color evenness was significantly different between strains (Kruskall–Wallis, *p* = 0.001). Post hoc tests indicated that GL values were significantly lower than TV (*p* = 0.01861) and UHM (*p* = 0.01521). Appendix A (Appendix A) illustrates the difference in material pigmentation for two materials under similar conditions after four (left) and three (right) weeks of growth.

Figure 3 shows the relationship between material’s colour homogeneity and its variance. 

### 3.3. Hydrophobicity

Figure 4 displays boxplots of the hydrophobicity measurements on materials from eight fungal strains. Hydrophobicity is expressed as the dilution of ethanol where the droplet was taken up by the material. A higher value, i.e., a higher concentration of ethanol that is absorbed by the surface, indicates a more hydrophobic material. Significant differences were observed between strains (ANOVA, *p* < 2.2 × 10^−16^) and between different strains using Duncan’s new multiple range test (results indicated by letters).

There is a significant positive correlation between hydrophobicity and the mean colorlessness (expressed as colorless ratio) of mycelial composites (Spearman’s correlation test, *p* = 2.249 × 10^−11^, rho = 0.7228195) (Figure 5).

### 3.4. Contamination Resistance

Figure 6 displays the ratio of the material surface covered by visible *Trichoderma* spores by strain and condition. For experiment A (BA, GA, GL, HE, PI, UHM), the influence of strain on the ratio of the material surface covered in visible *Trichoderma* spores (ratio *Trichoderma*) in the control group (C) (Kruskall–Wallis, *p* = 0.002055) is significant. In the treatment (T) (Kruskall–Wallis, *p* = 0.154) group, no significant effect was observed. There was a significant influence of fungal strain in both the C (Kruskall–Wallis, *p* = 0.0219) and the T (Kruskall–Wallis, *p* = 0.01672) for experiment B. See Appendix A for observed differences between strains. Within strains, a significant effect of treatment (CO (uncolonized fibers), C (control), or T (treatment)) was observed for all strains except for DT (Kruskall–Wallis test). More specifically, a difference between C and CO for BA (Dunn’s test, *p* = 0.00131), HE (Dunn’s test, *p* = 0.01022), UHM (Dunn’s test, *p* = 0.003933), and SH (Dunn’s test, *p* = 0.006235), a significant difference between CO and T for GA (Dunn’s test, *p* = 0.00561), GL (Dunn’s test, *p* = 0.00596), and HE (Dunn’s test, 0.0153), and a significant difference between C and T for TV (Dunn’s test, *p* = 0.0095913) were found. The Mann–Whitney–Wilcoxon test was used to test if the mean of CO for these aforementioned results was significantly larger than the mean of C and/or T, or if there was a significant difference between C and T, if the mean ratio *Trichoderma* was significantly higher in C than in T. See Appendix A for results.

The correlation between the mean colorless ratio of materials made from different strains and their variance was significant (Pearson’s correlation test, *p* = 0.0001397, rho = 0.94) (Figure 3).

## 4. Discussion

The values for thermal conductivity were quantified as the thickness of the conventional insulation material (in this case XPS) needed to achieve the same thermal conductivity as one centimeter of mycelial composite (cm XPS/cm material). The test was validated by using different thicknesses of XPS as a benchmark. We found that the test was both repeatable and accurate to predict thermal conductivity, as shown by the high and very stable Pearson’s correlation factors at each benchmark measurement iteration.

As seen in Figure 1 and Appendix A, the performance of mycelial composites in the insulation capacity test differed significantly between materials generated from different fungal strains. The estimated thermal conductivity of our samples was lower than what has been measured for mycelial composites in other studies (0.050–0.058 W/(mK) [23], 0.0882–0.104 W/mK) [22] or 0.0419–0.0578 [16]), with values below 0.037 W/(mK). Assuming a widely accepted thermal conductivity of 0.032–0.037 W/(mK) for XPS [3], the mycomaterials generated in our experiment have a lower thermal conductivity than XPS, some performing even twice better. This is comparable to, or even better than most commercially available alternatives (see Appendix A for a full comparison): glass wool, stone wool, PUR, and more sustainable insulation materials such as wood fibers (0.037–0.050 W/(mK), cork (0.037–0.050 W/(mK), and cellulose (0.037–0.042 W/mK) [3]. This phenomenon may be clarified by analyzing the density and porosity of the mycelial composites alongside their chemical composition. A lower overall density combined with an optimized ratio of micropores (which impede convective heat transfer) to macropores (which limit conductive heat transfer) could effectively account for the observed differences in thermal performance. The density values we measured (*D. tricolor*, *S. hirsutum*, *T. versicolor*) were between 113 and 150 kg/m^3^, indicating that the mycomaterials were significantly denser than hemp shives alone, suggesting that the improved insulation performance indeed lies in the porosity distribution. More importantly, this is lower than the thermal conductivity of pure hemp shives, 0.052 W/(mK) [15], meaning that fungal colonization improves the thermal conductivity of hemp shives. We speculate that fungal colonization and decomposition of hemp shives lead to the respiration of organic material, reducing substrate mass while maintaining its volume. This process increases the proportion of small pores within the material by enlarging void spaces as organic matter is removed. Simultaneously, hyphal growth within larger inter-shive pores decreases their size. As larger pores are more conducive to heat transfer via convection, this transformation results in a material structure with reduced thermal conductivity due to an overall increase in smaller, more insulating pores. Moreover, the properties of mycelial composites may be influenced by several other interrelated factors. First, the colonization density of the fungus is directly tied to its substrate preference, with fungi better adapted to the biochemical stoichiometry of hemp able to achieve denser growth. This adaptation also impacts metabolic activity, as the rate of substrate decomposition influences the material’s porosity by determining how much substrate is lost during growth, and therefore the distribution of pore size. The colonization pattern—whether the fungus grows predominantly within or on the surface of the substrate, generating a fungal skin—affects the composite’s structural properties. Lastly, the interconnectivity of the mycelial network plays a critical role; a higher number of cross-links within the network reduces pore interconnectivity, which in turn limits convective heat flow. Together, these factors shape the thermal, structural, and functional characteristics of the material.

Although not proven statistically, the materials with the least dense mycelial colonization (made with *F. fomentarius* and *G. lucidum*) also performed worse on thermal conductivity. It is rather logical that, as the fungal biomass itself acts as a binder between the particles of the hemp substrate, a denser hyphal network leads to a lower degree of connection between air pockets, and therefore a better resistance to heat transfer. Even though we did not design our experiment to test for this hypothesis, we speculate that the mycelial density likely reflects the ability of a given strain to exploit lignocellulosic material. Indeed, strains growing on lignin-based substrates usually grow to be denser and more hydrophobic [32]. The relationship between mycelial density and thermal conductivity might also be worth exploring as a function of sample thickness. Indeed, for samples exceeding approximately 10 cm in thickness, the fungal density decreases due to limited oxygen in inner layers, potentially reducing thermal conductivity.

Interestingly, there was sometimes a large variation in thermal conductivity within species. For example, with the *S. hirsutum* strain, one sample outperformed the XPS by a factor of two, whilst the thermal conductivity of another material was half as good. Even though growth conditions are uniform, this variation can be attributed to damage endured when removing the materials from their growth containers, the accuracy of the temperature sensors, or the small physical size of the samples, where minor stochastic heterogeneities within samples can have a disproportionately large influence that would be smoothed out in larger samples. This suggests that highly heterogeneous mycocomposites might also be more prone to lower structural strength, although we did not specifically test this property. Materials with low variances are preferred for production of mycelial composites because their material performance is more predictable. These results confirm the potential of mycomaterials as alternatives for conventional insulation materials [16,22].

We found that the materials differed significantly in their sensitivity to mold contamination, as seen in Figure 6. Materials that were most susceptible to mold attack are materials made with fungal strains where there was no significant improvement for the untreated group (C) compared to the uncolonized control fibers (CO). This was the case for *G. applanatum*, *G. lucidum*, and *p. nigricans* in experiment A, and for *T. versicolor* and *D. tricolor* in experiment B. Some strains do offer natural resistance to contamination, possibly due to the presence of antifungal metabolites. For example, *S. hirsutum* is known for its extensive biosynthesis machinery [33]. Interestingly however, we found that treating the material with deactivated mold spores while the mycelium was still developing in the substrate improved contamination resistance for several species (*G. applanatum*, *G. lucidum,* and *H. erinaceus*). This improvement likely resulted from the induction of antifungal metabolites by exposure to the deactivated spores. Three of these strains, *G. applanatum*, *G. lucidum,* and *T. versicolor*, were found to be susceptible when comparing the C and CO groups. Therefore, treating mycelial composites of these strains with deactivated *Trichoderma* spores during growth is recommended. The absence of any effect for treatment groups in other strains is not what we expected. Most strains, i.e., *D. tricolor*, *S. hirsutum,* and *T. versicolor*, have been shown to exhibit at least some (sometimes strain-dependent) antifungal properties by producing specific metabolites [34,35,36,37,38]. This would result in difficult-to-decompose fungal necromasses in the finished materials. In strains where the treatment group showed no effect, the limited exposure of the material to *Trichoderma* spores might be a contributing factor. This hypothesis is supported by the observation of more spores on the periphery of some materials and the absence of mold contamination at the exposure spot. *D. tricolor* and *P. nigricans* failed to improve microbial resistance in mycomaterials compared to the uncolonized hemp fibers, regardless of treatment, making them unsuitable for applications where microbial resistance is crucial, such as insulation panels.

A difference between C and CO, but not between T and CO, as observed in *B. adusta*, *Ganoderma spec.*, and *S. hirsutum*, could potentially be explained by several factors. These include biological variation, metabolites leached from deactivated spores damaging the growing mycelium, or stress factors influencing energy allocation during growth (resulting in less energy for antifungal metabolite production). Other possible explanations for the lack of effects across conditions includes the vaporization of metabolites during the drying process (70 °C), the fungus failing to produce metabolites in response to deactivated spores or to *Trichoderma* species in general, or the inefficacy of metabolites produced against the tested *Trichoderma* strain. We also feel the need to express that the strains we used were wild isolates and not domesticated strains, where the domestication process could have selected against these metabolites. Because the data for this test were pooled from two experiments, the growth conditions were not even, with *S. hirsutum*, *D. tricolor,* and *T. versicolor* growing on hemp fibers that were supplemented with 1% glucose and *B. adusta*, *G. applanatum*, *G. lucidum*, *P. nigricans*, *H. erinaceus*, and *Ganoderma spec*. growing on pure hemp fibers. This makes it impossible to make direct comparisons between these strains, but we speculate that the added glucose would not impact the results to the extent that making the general statement that fungal strain significantly influences resistance to contamination is unreasonable.

We also found that the color homogeneity (colorless ratio) of materials differed significantly between fungal strains, as shown in Figure 2 and Figure 3. Note that this index was independent from the color itself: *S. hirsutum* had, for example, a high color homogeneity while being ochre in tint. This was not surprising, considering the wide variability in the amount of pigments such as melanins [39] and terphenylquinones [40] across fungal species. We also found that color homogeneity was linked to colonization time. Samples exhibiting the highest color homogeneity had the lowest variance, aligning with expectations of fungal skin gradually transitioning from white to pigmented. Indeed, as the mycelium ages, condensation of water at the surface of the mycelium or nutrient depletion causes the affected surface to secrete pigments, resulting in a patchy color appearance (seen in Appendix A). The browning observed during growth may be attributed to exposure to radiation, fluctuations in environmental moisture, or nutrient depletion [41]. Considering that the materials with the most even coloration are more susceptible to having better perception, we would recommend the following species for generating insulation panels made of mycocomposites: *H. erinaceus*, *T. versicolor*, *Ganoderma* sp., and *S. hirsutum*.

We used ethanol dilution series displaying differential surface water tensions to determine hydrophobicity, as shown in Figure 4. All tested materials were hydrophobic (corresponding to contact angle values between 105 and 122° [30]), though at different levels depending on the fungal species. Mycelium surface hydrophobicity has already been found to differ between species. Our materials do not show the same variation in hydrophobicity as [42], where fungal mycelium from different soil fungal strains was either strongly hydrophilic or strongly hydrophobic. There was variation in hydrophobicity within materials, especially for the ones made with *P. nigricans* and *F. fomentarius*. A well-known class of molecules that functions in lowering water tension, and thus decreasing hydrophobicity on the surface of fungi, are the hydrophobins [43]. Differences in the usage and presence of hydrophobins and soluble pigments could explain the observed differences in hydrophobicity. The effect of pigmentation can be explained by examining the positive correlation between hydrophobicity and color evenness. More homogenous, non-pigmented materials were also more hydrophobic. This is consistent with what has been found in the literature for other fungi: higher degrees of hydrophobicity were observed on regions with white/aerial mycelia compared to older regions, with more soluble red pigments that replaced them in *Fusarium graminearum* [30]. This red pigmentation is related to melanin production and likely plays a role in desiccation stress [44] by storing water and ions, making the material more hydrophilic. Finally, the thickness of the fungal skin might also influence the overall hydrophobicity. The hydrophobicity of the materials can in turn influence its thermal conductivity by affecting how the material interacts with moisture. Hydrophobic materials resist water absorption, which is significant because water has a higher thermal conductivity than air. When a material absorbs water, the air-filled pores can be replaced with water, increasing heat transfer via conduction. Therefore, more hydrophobic materials tend to retain air within their structure and better insulate by minimizing the moisture-induced increase in thermal conductivity. This relationship is particularly relevant in moist or humid conditions. However, we did not observe a relationship between hydrophobicity and thermal conductance in our results (Spearman rho = 0.58, *p* = 0.11). We attribute this to the very narrow range of hydrophobicity of the tested materials (30% to 46% ethanol). The variation in hydrophobicity is therefore probably insufficient to detect a significant trend in its correlation with thermal performance.

This observation is consistent with our observation of replacement in time of white mycelial growth by the discoloration of the fungal skin and the positive relationship between whiteness and higher hydrophobicity. This has implications for mycomaterial research and production: to produce insulation panels that are more hydrophobic, and have more resistance in moisture uptake, it is vital to harvest materials before discoloration and avoid contact of aerial hyphae with water. The fungal skin exhibits critical characteristics, such as hydrophobicity and pigmentation, that could significantly influence key parameters relevant to insulation. These include enhancing thermal conductivity by mitigating water infiltration, which improves the material’s performance under varying environmental conditions.

## 5. Conclusions

We found that mycocomposites produced on hemp inoculated with nine different fungal strains had thermal conductivity comparable to the best synthetic insulation materials. All were hydrophobic, which is an advantage for materials exposed to many condensation cycles. Their appearance was very variable, but the ones more homogenous in color (and therefore with a potentially higher public acceptance) were also the more hydrophobic. We also found ways to improve their resistance to mold, which is a common issue with materials exposed to large variations in temperature and humidity. These findings highlight the importance of targeted selection of appropriate fungal strains with suitable traits to produce mycelial composites with desired characteristics. Further research into hydrophobin concentrations and the chemical nature of pigments produced might validate these hypotheses. Future research on mycomaterial insulation panels could focus on genetically engineering fungal strains to optimize mechanical, thermal, and hydrophobic properties. Additionally, exploring advanced composite fabrication techniques and integrating novel bio-based additives may enhance performance. Investigating the environmental durability and long-term insulation efficacy under diverse conditions also presents a critical avenue for further study. Further validation of these hypotheses can be achieved through screening materials for known antifungal metabolites and conducting molecular analysis of antifungal metabolite pathways in fungal strains. Additionally, investigating the long-term duration of the observed effects is necessary, as insulation panels are expected to last 10 to 20 years.

## Figures and Tables

**Figure 1 materials-17-06050-f001:**
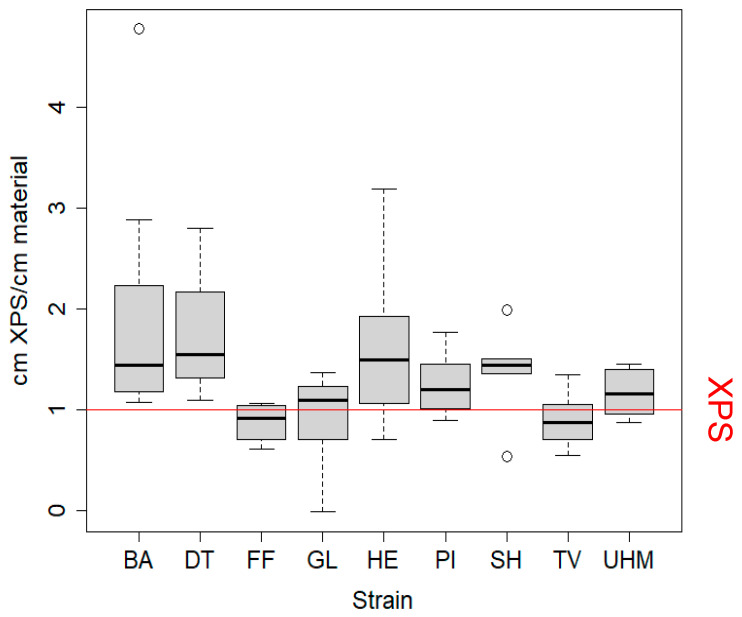
Thermal conductivity of mycomaterials was measured as equivalent XPS thickness needed per cm of material to achieve the same insulation effect. Values above 1 indicate that the mycomaterial outperformed XPS. Samples (7 cm diameter) were sealed in insulated boxes, preheated to 60 °C, and cooled for 30 min. Temperature differences were converted to equivalent XPS thickness and divided by sample thickness. Statistical analysis included Kruskall–Wallis (*p* = 0.006), *t*-tests, Mann–Whitney–Wilcoxon, and Dunn’s tests, though the latter showed no significant strain differences. Samples were prepared using method A to evaluate strain effects on thermal performance. BA = *Bjerkandera adusta* (n = 8), DT = *Daedaleopsis tricolor* (n = 4), FF = *Fomes fomentarius* (n = 4), GL = *Ganoderma luciducm* (n = 8), HE = *Hericium erinaceus* (n = 8), PI = *Phellinus nigricans* (n = 6), SH = *Stereum hirsutum* (n = 5), TV = *Trametes versicolor* (n = 8), UHM = *Ganoderma spec* (n = 8).

**Figure 2 materials-17-06050-f002:**
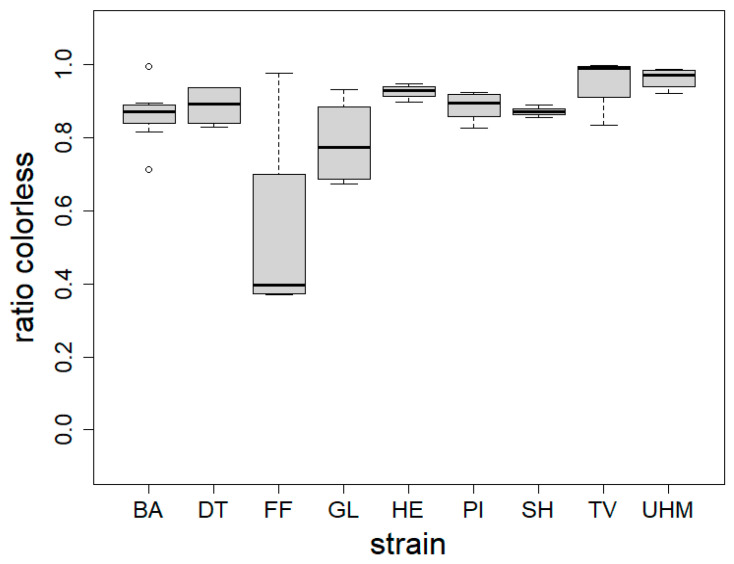
Material color homogeneity (colorless ratio) expressed as the ratio of photographed material surface that is brighter than bright red after conversion to 8-bit grayscale for each fungal strain (strain). BA = *Bjerkandera adusta* (n = 8), DT = *Daedaleopsis tricolor* (n = 4), FF = *Fomes fomentarius* (n = 4), GL = *Ganoderma luciducm* (n = 8), HE = *Hericium erinaceus* (n = 8), PI = *Phellinus nigricans* (n = 6), SH = *Stereum hirsutum* (n = 5), TV = *Trametes versicolor* (n = 8), UHM = *Ganoderma spec* (n = 8). Samples were generated following method A. The Kruskall–Wallis test was used to determine the influence of fungal strain on colorlessness (*p* = 0.001), Dunn’s test was used for post hoc analysis: GL differed from TV (*p* = 0.0186) and UHM (*p* = 0.015).

**Figure 3 materials-17-06050-f003:**
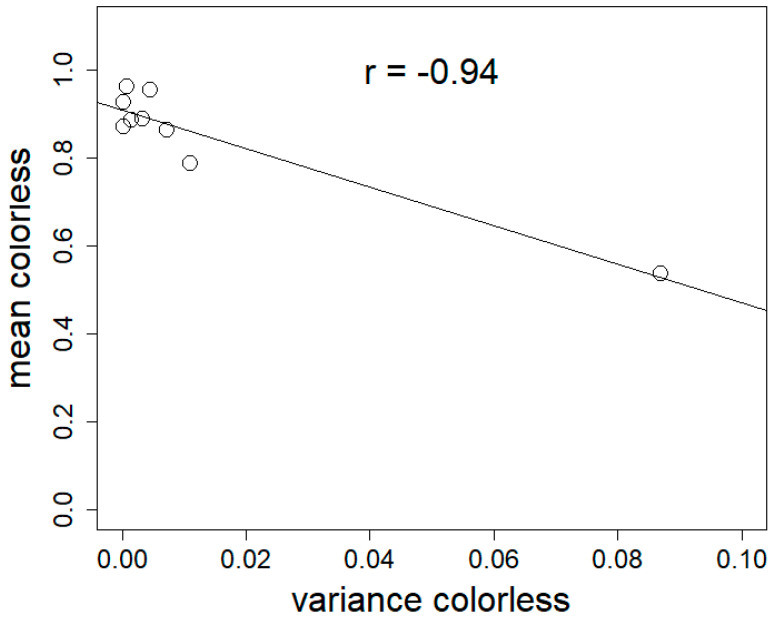
Mean colorless ratio (mean colorless) as a function of the variance in colorless ratio (variance colorless) for materials made with different fungal strains. Linear correlation confirmed by Spearman’s correlation test (*p* = 0.0001397), rho = −0.94.

**Figure 4 materials-17-06050-f004:**
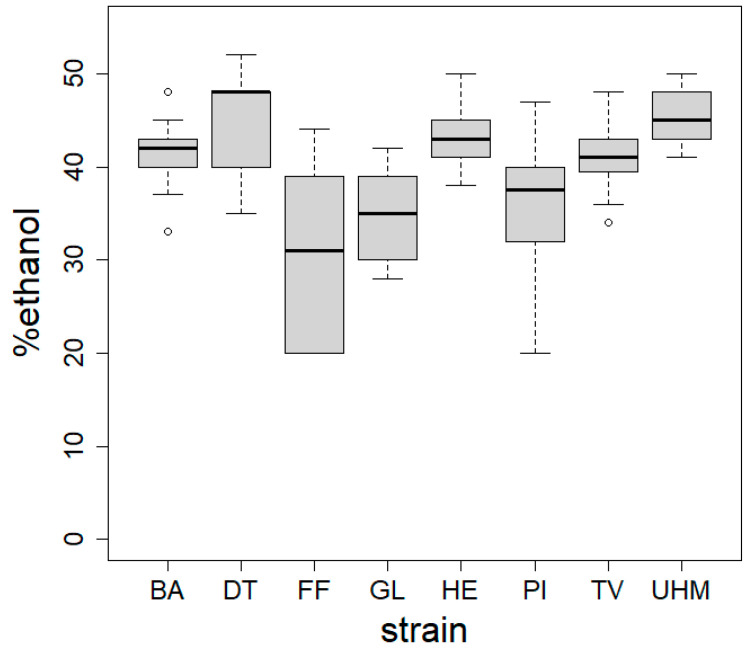
Hydrophobicity of mycelial composites assessed using the ethanol percentage test across five spots per sample. Materials were produced via method A. ANOVA revealed a significant strain effect on hydrophobicity (*p* < 2.2 × 10^−16^), with Duncan’s test identifying inter-strain differences. BA = *Bjerkandera adusta* (n = 8), DT = *Daedaleopsis tricolor* (n = 4), FF = *Fomes fomentarius* (n = 4), GL = *Ganoderma luciducm* (n = 8), HE = *Hericium erinaceus* (n = 8), PI = *Phellinus nigricans* (n = 6), SH = *Stereum hirsutum* (n = 5), TV = *Trametes versicolor* (n = 8), UHM = *Ganoderma spec* (n = 8).

**Figure 5 materials-17-06050-f005:**
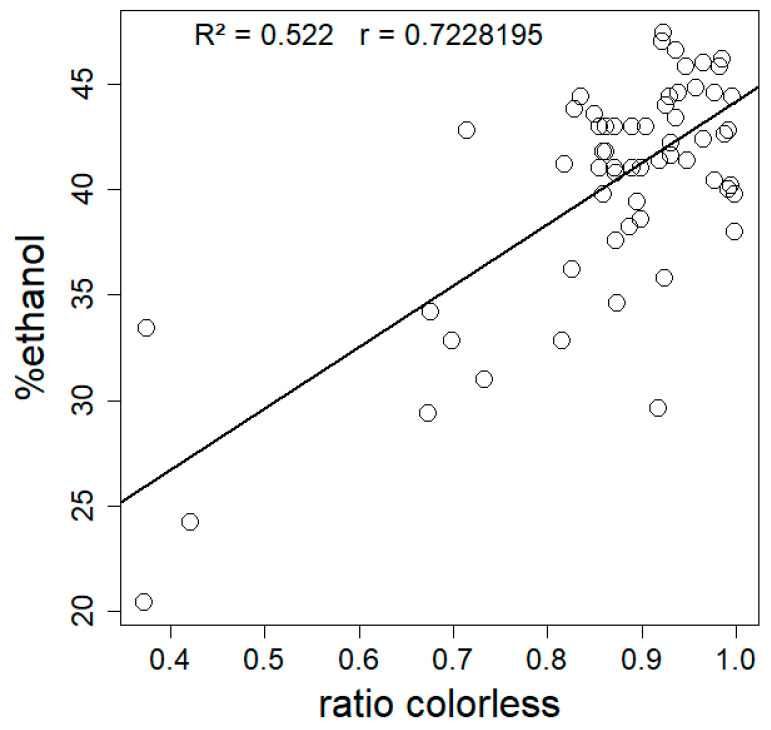
Mean hydrophobicity (n = 5) expressed as %ethanol as a function of mean colorless ratio (whiteness) for 72 composites made from nine different fungal strains. Statistical analysis by Spearman’s correlation test (rho = 0.7228195, *p* = 2.249 × 10^−11^), R2 of the linear model was 0.522.

**Figure 6 materials-17-06050-f006:**
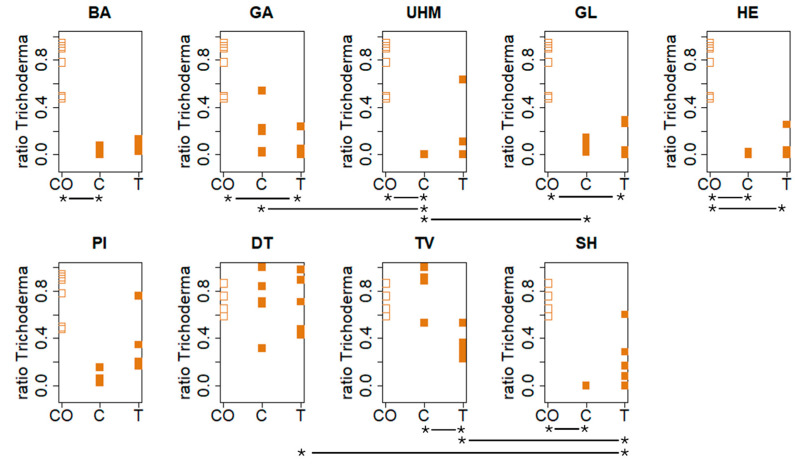
Visible *Trichoderma* spore coverage was analyzed on material surfaces under different conditions (CO: control with only substrate not inoculated with the tested strain, T: deactivated spores, and C: untreated) six days post-exposure. Materials were incubated at 30 °C and 90% RH after fungal inoculation. Statistical analyses (Kruskall–Wallis, Dunn’s, and Mann–Whitney–Wilcoxon tests) identified significant effects of fungal strain and treatment group on spore ratio, with inter-strain and group differences varying by experiment and condition. The “*--*” highlighs significant differences in contamination resistance between two treatments for each strain. See the results section and Appendix A for specific strain comparisons and *p*-values.

## Data Availability

The original data presented in the study are openly available in Dryad at 10.5061/dryad.zpc866tjr.

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
