# Peer review of "Fungal Strain Influences Thermal Conductivity, Hydrophobicity, Color Homogeneity, and Mold Contamination of Mycelial Composites"

_materials, 2024, doi:10.3390/ma17246050_

Round 1
Reviewer 1 Report
Comments and Suggestions for Authors
The research explores how different fungal strains influence key properties of mycelial composites, including thermal performance, hydrophobicity, color homogeneity, and biodegradability. It highlights the potential of strain-specific optimization to tailor these composites for diverse applications in sustainable materials science. The topic is significantly important but there some major concerns which needs to be addressed before journal make decision. My comments are given below-
Fungal Strain's Role: The paper’s focus on the fungal strain's impact on material properties is both novel and critical. However, a deeper exploration into the genetic and physiological mechanisms by which different fungal strains influence these properties would strengthen the conclusions.
Hydrophobicity Metrics: The discussion on hydrophobicity could benefit from quantitative data, such as contact angle measurements, to substantiate claims regarding performance differences between fungal strains.
Thermal Performance and Standardization of Tests: The study mentions "thermal performance," but it is unclear if this refers to thermal conductivity, stability, or decomposition temperature. Providing explicit metrics and comparisons with alternative materials is essential for broader application insights. The variability inherent in biological systems like fungal mycelium underscores the need for standardized testing protocols. Were the tests conducted under consistent environmental conditions? This should be clarified to ensure reproducibility.
Biodegradability Analysis: Biodegradability is an important factor, yet the time scale of degradation and the environmental conditions under which tests were performed are not adequately detailed. Including this would provide a clearer picture of the practical implications.
Structural Homogeneity in Application-Specific Context: The paper notes color and structural homogeneity but could discuss how these aspects affect mechanical and aesthetic properties. For example, how does heterogeneity influence mechanical strength or potential market applications? Also, cite DOI: 10.2174/0115734110316443240725051037 report with the sentence ending with the ´… absorbed by the material surface (dependent variable), was determined using ANOVA´ to make references up to date.
The research would be more impactful if it explicitly addressed potential application contexts, such as packaging, construction, or biodegradable plastics, and the specific material requirements for these fields.
Comparative Analysis: While the fungal strain’s influence is highlighted, the work would benefit from a broader comparison to other biomaterials, such as bacterial cellulose or synthetic biopolymers, to underscore the unique advantages or limitations of mycelial composites.
Processing Parameters: Detailed information on the cultivation and post-processing conditions (e.g., humidity, substrate type, or drying process) is crucial for replicability and optimization of material properties.
Environmental Impact: A lifecycle assessment (LCA) or discussion on the environmental trade-offs of producing mycelial composites with different fungal strains would strengthen the sustainability argument.
Scalability and Interdisciplinary Context: The scalability of producing mycelial composites with strain-specific properties is a significant factor for industrial applications but remains underexplored. Addressing this would enhance the paper’s practical relevance. Incorporating interdisciplinary insights, such as microbiology or material science, would deepen the discussion of strain-specific properties and their translation into material performance. In addition, with tzhe sentence ´ The colonization by mycelium significantly improves the physico-chemical properties of the material. ´cite https://doi.org/10.1007/s44174-024-00206-z to update the literature.
Characterization Techniques and Future Research Directions: The chosen methods for evaluating properties like biodegradability and thermal performance are not detailed. Including specifics about characterization techniques (e.g., thermogravimetric analysis, FTIR) would enhance credibility. While the paper identifies strain variability as a key factor, it could more explicitly outline future research areas, such as genetic engineering of fungal strains for targeted property optimization.

Reviewer 2 Report
Comments and Suggestions for Authors
The manuscript is interesting, and the results and discussions are satisfactory. However, the following comments should be considered further improvements to the manuscript.
1. Unify the unit of thermal conductivity throughout the manuscript.
2. It would be beneficial for the reader's understanding if the authors included a schematic diagram in the section "2.2. Preparation of composite materials."
3. Revise it clearly “The smaller the increase, the better the thermal performance.”
4. It would be better if the authors moved Sections 2.3, 2.4, 2.5, 2.6, and 2.7 to supplementary files.
5. The supplementary file should include the author's details and affiliation.
6. Section 3.1 should discuss the purpose of the thermal performance.
7. Why does Fig. 1's caption have such a lengthy paragraph? It should be simple and easy to understand in the background of the figure.
8. Similarly, the captions for Fig. 4 and Fig. 6 may be simplified to better explain the background of the figures.
9. Why do mycomaterials have poorer thermal conductivity than XPS and pure hemp shives? It should be addressed on page 12.
10. The discussion on page 12 could connected to the corresponding Figures.
11. Line 451 should include the density of the mycelium network for achieving lower thermal resistance.
12. Check the spelling of "Whiel" on Line 472.
13. Why are lines 486–488 alone? Connect with the relevant sentences.
14. Revise lines 489-493 due to the long phrase.
15. Similarly, lines 494-499 are long. Revise it two or three sentences.
16. Why some species have grown on pure hemp fibers while others are not is explained in lines 502 to 504. Discuss it.
17. What is the rationale for the highest color homogeneity with the lowest variance?
18. If possible, the supplemental data should relate to the appropriate statement in the discussion section.
19. It would be better if the authors addressed the challenges and future scope of the current study in the conclusion. In addition, simplify the conclusion since it seems to be long discussion.
Reviewer 3 Report
Comments and Suggestions for Authors
The authors report on the investigation titled “Fungal Strain Influences Thermal Performance, Hydrophobicity, Color Homogeneity and Mold Biodegradability of Mycelial Composites.” The study appears to be well-conducted and the results are presented clearly. The manuscript may be considered for publication in this reputed journal with the following revisions:
Need to rewrite the abstract with numerical results of Thermal performance, hydrophobicity, color homogeneity and mold biodegradability of Mycelial Composites and highlight the contribution of the present study. Please avoid the general information in the abstract.
The introduction is excessively lengthy, resembling that of a review article or dissertation. In comparison to the results and discussion section, the introduction is disproportionately extensive in this study. Also, authors need to state the clear novelty of the present work in the introduction.
Why did the authors not provide morphology analysis of nine fungal strain? The strain morphology will affect the thermal performance, colorlessness, and hydrophobicity. So, the authors need to provide the morphological analysis of fungal strains and correlated with thermal performance, colorlessness, and hydrophobicity properties in the text.
Any correlation between thermal performance and hydrophobicity of fungal strains. Explain in detail.
Role of sample thickness in thermal performance, colorlessness, and hydrophobicity of mycomaterials. If change the circular cutout diameter in the thermal performance measurement (Figure 1), what will happened in thermal performance of mycomaterials? Explain it.
Authors need to provide the estimated thermal conductivity of prepared samples in a tabular format. So that audience easy to understand the performance of mycelial composites in the insulation capacity test.
Suggest a possible mechanism for the influence fungal strains on thermal performance, hydrophobicity, color homogeneity and mold biodegradability of mycelial composites.
Conclusion seems like summary. The authors should rewrite the conclusion with present study highlighted results and future perspectives.
Researchers can present their findings alongside comparable studies in a table to emphasize the strengths of their research.
Sometimes the authors used italic format for fungal strains and other times used without italic format for fungal strains. Unify them throughout the text and especially in conclusion section.
Please always leave a space between the number and SI unit (e.g., 150 nm, 21 °C, 5 h, etc.), however no space before the "%", "/" and ":" signs. Please check all units in the whole text. There are inconsistencies in the expression and format of units.
Too much typographical errors are observed in the present manuscript and also improve the English language of the manuscript.
Round 2
Reviewer 1 Report
Comments and Suggestions for Authors
accept